# Analysis of the Mechanical Properties of 3D-Printed Plastic Samples Subjected to Selected Degradation Effects

**DOI:** 10.3390/ma16083268

**Published:** 2023-04-21

**Authors:** Josef Sedlak, Zdenek Joska, Jiri Jansky, Jan Zouhar, Stepan Kolomy, Martin Slany, Adam Svasta, Jan Jirousek

**Affiliations:** 1Faculty of Mechanical Engineering, Brno University of Technology, 601 90 Brno, Czech Republic; sedlak@fme.vutbr.cz (J.S.);; 2Department of Mechanical Engineering, Faculty of Military Technology, University of Defense in Brno, 662 10 Brno, Czech Republic; 3Department of Mathematics and Physics, Faculty of Military Technology, University of Defense in Brno, 662 10 Brno, Czech Republic; 4Department of Combat and Special Vehicles, Faculty of Military Technology, University of Defense in Brno, 662 10 Brno, Czech Republic

**Keywords:** 3D printing, FFF technology, aging, degradation factors, tensile test, hardness test

## Abstract

The Fused Filament Fabrication (FFF) method is an additive technology that is used for the creation of prototypes within Rapid Prototyping (RP) as well as for the creation of final components in piece or small-series production. The possibility of using FFF technology in the creation of final products requires knowledge of the properties of the material and, at the same time, how these properties change due to degradation effects. In this study, the mechanical properties of the selected materials (PLA, PETG, ABS, and ASA) were tested in their non-degenerate state and after exposure of the samples to the selected degradation factors. For the analysis, which was carried out by the tensile test and the Shore D hardness test, samples of normalized shape were prepared. The effects of UV radiation, high temperature environments, high humidity environments, temperature cycles, and exposure to weather conditions were monitored. The parameters obtained from the tests (tensile strength and Shore D hardness) were statistically evaluated, and the influence of degradation factors on the properties of individual materials was assessed. The results showed that even between individual manufacturers of the same filament there are differences, both in the mechanical properties and in the behavior of the material after exposure to degradation effects.

## 1. Introduction

Fused Deposition Modeling (FDM) and Fused Filament Fabrication (FFM) refer to the same additive manufacturing process, where a thermoplastic material is melted and deposited layer by layer to create a three-dimensional object. FDM is a trademarked term, while FFM is a generic term used by some manufacturers to describe the same process without using the trademarked name. This type of additive technology is the most widespread method due to the affordability of production equipment and materials. The input material is in the form of wire. The principle of the method is the melting of the input material in the print head and its layering into the desired shape [1]. The wire, or filament, is fed into the print head—the extruder. The printhead consists of a feed mechanism (stepper motor and feed wheels) and a hot end (an assembly of parts that are used to fuse and lay the material through the nozzle). The contour of laying the material consists of outer loops (perimeters) and inner filling, which can have a different shape and density. When creating parts with a more complex shape, the construction of supporting material is necessary. FFF technology is used for the creation of prototypes and final products in a number of industries [2]. Dey et al. [3] investigated the performance, limitations, and opportunities of different types of FFF fibers, which may contain different types of reinforcements such as particles, fibers, and nanoparticles that improve the properties of FFF-printed components. FFF technology is primarily used to process plastic materials; PLA (Polylactic Acid), PETG (Polyethylene Terephthalate Glycol), ABS (Acrylonitrile Butadiene Styrene), and ASA (Acrylonitrile Styrene Acrylate) materials are currently some of the most commonly used [2]. The mixed recycled material and the original material improve the mechanical properties of the recycled fiber even after five recycling cycles [4].

Woern et al. [5] investigated the suitability of recyclable materials for 3D printing on Gigabot X. This printing achieved the printing of large amounts of polymers in less time and reduced the cost of plastic components, as pellets and recycled plastic are cheaper input materials. In studies focused on 3D printing, most authors focus on the tensile strength of the material [6,7,8]. The flexural strength of a PLA part made of fused fiber (FFF) is primarily influenced by the layer height and the consequent raster angle [9]. In addition to the layer height and screen angle, the influence of other process parameters (fiber thickness, orientation, screen width, and air gap) on tensile strength, flexural strength, and the impact of the sample during 3D printing were also investigated [10]. Bonada et al. [11] draw attention to the necessary knowledge of the mechanical behavior of printed samples depending on the parameters of 3D printing. The results of this study point to differences in the mechanical properties of FFF-printed parts depending on the internal arrangement of the fill pattern under similar 3D printing conditions. Polylactic acid (PLA) is the most widely used material used for 3D printing; it is a degradable polymer with a chemical synthesis of acetaldehyde and through the carbohydrate fermentation process [12,13]. Breški et al. [14], in their study focused on recycled PLA, state that infill density has almost no influence on tensile strength, or even a negative influence in the case of minimum layer height. Polylactic acid (PLA) vascular stents with a negative Poisson’s ratio (NPR) structure were made by 3D printing. Compression experiments showed that wall thickness had the greatest effect on the mechanical properties of PLA stents [15]. The change in the mechanical properties of the PLA material under the influence of a temperature of 60 °C and an environment with increased humidity was monitored by the study of [16], who reported there was a decrease in the modulus of elasticity in tension, the more pronounced the higher the humidity of the environment. In a similar way, there was also a drop in the ultimate strength, while the ductility, on the contrary, increased with increasing humidity. A decrease in ultimate strength and modulus of elasticity in tension due to the effect of moisture on PLA was also observed in the study of [17]. The effect of UV radiation and elevated temperature on the mechanical properties of the PLA material is described in the study of [18], which results in a decrease in the tensile modulus and ultimate strength due to temperature and exposure to UV radiation. Woern et al. [5] investigated the suitability of recyclable materials for 3D printing on Gigabot X. This printing achieved the printing of large amounts of polymers in less time and reduced the cost of plastic components, as pellets and recycled plastic are cheaper input materials. In studies focused on 3D printing, most authors focus on the tensile strength of the material [6,7,8]. Yonezawa and Yamada [19] report the deteriorating mechanical properties of PLA test pieces prepared on a 3D FFF printer, which were immersed in saline and evaluated by tensile and bending tests. In the case of PETG material, an increase in the tensile strength was observed as the time of the experiment increased due to the effect of elevated temperatures below the glass transition temperature [20]. The trend of improvement in the mechanical properties of PETG is also confirmed by the study of [21], during which the samples were exposed to the outdoor environment and the yield strength and modulus of elasticity in tension increased. Adding carbon fibers to polymeric composite materials (PETG) will significantly reduce the coefficient of thermal expansion, further supporting the claim that composites made of cheaper recyclable PETG have comparable properties to the original PETG composites [22]. Duralek et al. [23] emphasize the reuse of printed films made from glycol-modified PETG, and the results of the study point to comparable tensile strength. The mechanical properties of ABS and ASA also improved at elevated temperatures below the glass transition temperature [24]. The authors of [25] describe the influence of the mechanical properties of ABS material due to the action of a humid environment. After immersing the samples in water heated to 50 °C, the tensile strength modulus decreased, the ductility of the material increased, and the value of the strength limit was not significantly affected. The effect of UV radiation on ABS material was observed in the study of [26]. Extruded ABS samples were exposed to UV radiation for various lengths of time, while the tensile strength and modulus of elasticity decreased. The resistance of ABS and ASA to the effects of increased temperature, UV radiation, and environments with increased humidity can be compared on the basis of the research in article [27]. Ductility decreased in ABS samples and, to a lesser extent, in samples with a covering layer of ASA material. The tensile strength of the ABS samples decreased, but the combination of ABS + ASA was not affected. The tensile modulus values did not change significantly. The effect of FT temperature cycles (Freeze-Thaw Cycles—temperature cycles of frost/room (elevated) temperature) on the selected materials can be estimated based on the behavior of other polymers. The effect of FT cycling on FDM 3D printed samples made from a mixture of polyetherimide and polycarbonate is described in the study of [28], where cycling caused a decrease in tensile strength and modulus of elasticity. A decrease in ultimate strength was also observed for the DGEBA epoxy resin, where ductility also decreased and the modulus of elasticity in tension, on the contrary, increased [29]. For composites of HDPE (high-density polyethylene) and PP (polypropylene) with wood flour, a decrease in yield strength and modulus of elasticity in tension was observed [30]. For the use of FFF technology to create final components, knowledge of the mechanical properties of the material is necessary, and at the same time, the change in mechanical properties due to the action of degradation factors affects the choice of material for specific conditions of use. The purpose of this study is to determine the influence of degradation factors on the mechanical properties of samples prepared by FFF technology. Determination of mechanical properties was performed for PLA, PETG, ABS, and ASA materials in as-printed states and after exposure to degradation factors. The benefit and novelty of this work lies in the combination of degradation tests, when both standard degradation tests and fewer common tests were included in the experiment. The results of these tests will help the 3D printer community select the appropriate material for possible applications.

## 2. Experimental Setup and Material

### 2.1. Material—Shape of Samples

The shape and dimensions (in mm) of the samples seen in Figure 1 for the tensile test were determined based on the standard EN ISO 527-2:2012, Plastics—Determination of tensile properties—Part 2: Test conditions for molded plastics [31]. Although this standard is not intended for 3D printed plastics, given that it is most often used in the literature together with the ASTM D638 Type-I standard, it was used in this experiment [32,33].

In order to save material, reduce costs, and reduce production time, the shape of the 1BA samples was fabricated for testing ultimate tensile strength [31].

The test bar model was created in CATIA v5-3DX parametric CAD software and exported in .stl format. The samples met the requirement of a minimum thickness of 4 mm of the test body according to the standard EN ISO 868, Plastics and ebonite—Determination of hardness by indenting the tip of a hardness tester (Shore hardness) [34]. Therefore, for hardness measurement, the Shore D method was performed. Places where the hardness was evaluated are depicted in Figure 2.

Materials such as PLA, PETG, ASA, and ABS were chosen to produce test samples. Products from three different manufacturers were chosen for each material (see Table 1). For each type of material and manufacturer, 7 test samples were produced for each series of testing.

PLA is a thermoplastic made from sugar cane, potato, or corn starch. The advantage of PLA is that it is a material produced from renewable resources that is in some way biodegradable, unlike other thermoplastics produced from petroleum. Another advantage of PLA is that it is easy to print, thanks to its low melting point and low thermal expansion (see Table 2), and there is no print shrinkage. Therefore, it can be used for larger prints and more detailed models. The disadvantage of PLA is its low glass transition temperature, which makes it unsuitable for use at higher temperatures or for machining (without cooling) [35,36,37].

ABS is an amorphous thermoplastic copolymer that was one of the first materials used for FFF/FDM technology. Its advantages are good mechanical properties and resistance to high temperatures. ABS is soluble in acetone, which can be used for gluing or smoothing layers. The disadvantages of ABS material are its low resistance to UV radiation and large shrinkage due to thermal expansion, which makes ABS unsuitable for printing larger components. To reduce shrinkage, it is advisable to use heated chambers (FDM) when printing it. When printing ABS, styrene is released, which creates an unpleasant smell, so it is advisable to process it in a ventilated room [35,36,37].

Thermoplastic PETG is a modification of one of the most widely used plastics, PET (Polyethylene-Terephthalate). The modification consists of the addition of modified glycol during polymerization, which increases the toughness of the material. The advantages of PETG are its higher elasticity and higher temperature resistance compared to ABS material. Another advantage compared to ABS is its lower thermal expansion, so PETG can also be used for printing larger objects [35,36,37].

ASA is an amorphous thermoplastic that has similar properties to ABS (see Table 2), but its advantages are lower shrinkage and higher resistance to UV radiation. ASA is a suitable material for applications where appearance is emphasized, as it does not yellow when exposed to UV radiation. Like ABS, a harmful smell is released when printing ASA, so it is advisable to print it in a ventilated room [35,36,37].

### 2.2. Printers Used and Printing Parameters

The model in *.stl format was imported into the PrusaSlicer program, which is used to generate the control program of the 3D printer. The following print parameters were selected:Layer thickness: 0.15 mm (first layer: 0.2 mm);Three perimeters;Straight, 100% filling, oriented alternately at 45° and −45°;Orientation of the samples: flat on the mat and in a horizontal position;Nozzle 0.4 mm;Printing parameters and speed for selected materials are shown in Table 3.

The printer control program was subsequently exported from the PrusaSlicer 2.3.5 software in .gcode format.

Original Prusa i3 MK2S and Original Prusa i3 MK3S FFF printers from the manufacturer Prusa Research were used for the production of the samples; see Figure 3. These are devices belonging to the hobby category that achieve lower accuracy levels compared to professional printers, but their main advantage is affordability.

## 3. Exposure of Samples to Degradation Influences

The samples produced were subsequently exposed to 4 different degradation effects, when the effects of humidity, temperature, UV radiation, and winter weather were investigated on the main mechanical properties of the material. The characteristics of individual degradation effects are presented below.

### 3.1. Condensation Chamber

A set of samples for placement in the condensation chamber, see Figure 4, were weighed on a digital laboratory scale (Citizen Scale CY 720) before being placed in the chamber, so that the amount of absorbed moisture could be estimated based on the change in the weight of the samples before and after being placed in the chamber, see Table 3. The condensing chamber KB300 CONSTANZO was used for testing the samples. The samples were exposed to an environment of 100% humidity and a temperature of 55 °C for 100 h in the condensation chamber.

The average percentage increases in weight of individual materials after 100 h in the condensation chamber are shown in Table 4.

### 3.2. Temperature Cycles

Temperature cycling took place by alternating exposure to frost at −18 °C (±1 °C) and room temperature at 21 °C (±2 °C). The samples were subjected to a total of 130 such cycles, with the duration of one cycle being 16 h. The course of the temperature cycle is shown in Figure 5.

### 3.3. UV Radiation

UV irradiation of the samples with a Mikrolux Chirana device with a mercury discharge lamp with a power of 125 W took place at 400 mm/20 h and 100 h, respectively. The samples were rotated at regular intervals so that the illumination from both sides was uniform.

After 20 h, color changes could be observed on the ABS samples (for lighter shades), showing a slight yellowing, and the surface of the PLA samples became sticky. No visual changes were observed in the PETG and ASA samples. The second set was illuminated for 100 h; after this time, the color changes were more noticeable in the ABS material samples, e.g., Fillamentum ABS (see Figure 6), and were also slightly evident in the ASA material samples.

### 3.4. Endurance at an Elevated Temperature

Another set of samples was placed in the furnace for 100 h at a temperature of 60 °C (see Figure 7). This temperature was chosen because there would be no significant shape deformation of the samples and thus it would be possible to test their mechanical properties. At the same time, this is the glass transition temperature range for PLA material. For the remaining materials, this is the temperature below the glass transition temperature.

### 3.5. Weather Effects

The last method of aging the samples was their placement in an outdoor environment for 98 days. The samples were placed in a covered area and were simultaneously affected by changes in temperature, sunlight, and air humidity. Data from an amateur meteorological station [41], less than 7 km as the crow flies from the testing site, were used to document the development of the weather. The air temperature fluctuates between −5 at night and 10 °C during the day. The air humidity was between 30 and 97%. The samples were subjected to aging for 2400 h. The total length of sunshine for the entire testing period can be estimated at 294 h, which amounts to an average of 3 h per day. Only solar radiation with a minimum intensity of 120 W·m^−2^ is included in the length of sunshine [41].

## 4. Testing of Mechanical Properties

A Zwick Z100 testing device was used to perform the tensile test. The tests were carried out according to the EN ISO 527-2 standard, and the monitored parameters were ultimate tensile strength and hardness.

The hardness was measured by the Shore D method, and the Bareiss Digi-Test II hardness tester was used for the measurement, according to the EN ISO 868 standard. The ends of the test rods were selected for the measurement.

Figure 8, Figure 9, Figure 10 and Figure 11 show representative graphs of selected representatives of individual types of materials, where the curves during the tensile test are shown. These curves show the changes caused by individual degradation effects. In Section 6, the achieved results are described in more detail, based on the results of the statistical analysis, where individual types of material are compared with each other. Tensile diagrams of the remaining materials are in the Appendix A.

## 5. Statistical Evaluation of Results

Excel software and MATLAB 9.5 mathematical software were used for data processing. The first step of the statistical processing was the normality test, which is a prerequisite for the calculation of statistical parameters. Due to the number of samples, the Anderson–Darling normality test was used, which states that if the *p*-value is greater than 5%, the data are more than 95% normally distributed. Since the compared data had statistically significantly different variances, their comparison was performed using the Kruskal–Wallis test.

The evaluated parameters (Table 4 and Table 5) were further compared with reference values to evaluate the influence of individual degradation factors. The ANOVA test was used to compare the values, based on which it is possible to declare a 95% confidence level of whether the values are identical.

## 6. Results and Discussion

### 6.1. Comparison of Results between Producers

Statistical analysis was performed on the results of ultimate tensile strength tests and hardness tests.

The properties of PLA and PETG materials (see Table 5 and Table 6) were almost independent of the manufacturer. Both materials showed, on average, identical ultimate tensile strength limits, given the effect of wear. Furthermore, the measured tensile strength values were, given the effect of wear, higher than for the other two studied materials. In terms of hardness, PLA materials showed slightly higher hardness than all other materials studied. On the contrary, the PETG materials showed a slightly lower hardness on average than the other studied materials.

For ASA and ABS materials, see Table 7 and Table 8. Differences between manufacturers have already been noted. For the ASA material, samples from Plasty Mladeč showed a higher tensile strength limit than those of the other two manufacturers. Additionally, this is both for the reference sample and for its influences. There were no significant differences between manufacturers in the hardness values of this material. For the ABS material, Gembird samples showed higher ultimate tensile strength and often higher hardness than samples from the other two manufacturers.

### 6.2. Evaluation of Individual Influences

Statistical analysis was performed again for hardness and ultimate tensile strength. The impact of individual influences on the sample was evaluated using the Kruskal–Wallis test.

After exposure to the effects of UV radiation, the ultimate tensile strength decreased slightly, but statistically significantly, only for the manufacturer, Plasty Mladeč, of the PLA material. For the other materials, the assumption of a drop in ultimate tensile strength was not confirmed. The hardness of the materials was surprisingly increased due to the action of the “UV Lamp” in PETG materials produced by Plasty Mladeč, ABS materials produced by Fillamentum and Gembird, and ASA materials produced by C-TECH. It did not change statistically significantly for other manufacturers.

After exposure to “UV 100 h”, there was a slight increase in the tensile strength of PLA and PETG plastics. For other types of plastic, the tensile strength limit has hardly changed. The reason may be insufficient intensity or low irradiation time. The hardness of PLA and ASA plastics decreased slightly when exposed to “UV 100 h”. For PETG and ABS plastics, changes in hardness were statistically insignificant, except for the Spectrum PLA plastic manufacturer, where the hardness increased slightly.

After the samples were exposed to the influence of the condensation chamber environment, there was a drop in tensile strength only for the PLA material from the manufacturer Plasty Mladeč and for the ABS material from the manufacturer Fillamentum. In other cases, there was either no significant influence or, on the contrary, in the case of PETG plastic, a slight increase. There is a slight increase in modulus of elasticity, except for the PLA material. Because of water absorption, an increase in ductility was expected on the basis of the research, but this was not confirmed—a slight increase occurred only in the case of the PLA material; on the contrary, in the case of PETG, it decreased considerably. A decrease in or unchanged ductility values could also be observed for all other materials. In the case of hardness, it was possible to observe a decrease in the PLA material, but the hardness of the others was not affected. Therefore, some of the mechanical properties of the PLA material described in the articles [3,42] were not confirmed. For example, the authors of [3] state that the mechanical properties of PLA are poor, have a rough texture, and are brittle. Similarly, the authors of [43,44] state that PLA material has a low thermal conductivity and toughness, i.e., it shows lower deformation in an as-printed state.

The change in properties after exposure to temperature FT cycles could not be predicted due to the lack of similar studies for the applied materials. The ultimate tensile strength decreased slightly for the ABS material. On the contrary, ultimate tensile strength was not negatively affected for the PETG, ASA, and PLA materials or the manufacturers of these materials. Conversely, the hardness of ABS and PETG materials increased. For PLA and ASA plastics, the increase in hardness was not statistically significant due to (UV) damage to the polymers and polymer matrix. Due to the effect of “temperature cycles”, there were no statistically significant changes in the ultimate tensile strength or hardness of the material, with the exception of ABS Fillamentum plastic, where the tensile strength was reduced.

Because of “Resistance at elevated temperature”, the tensile strength and hardness limit of PETG plastic increased, and this was the case for all the manufacturers. ABS and ASA plastics increased only in hardness. For PLA plastic, there were no statistically significant changes in hardness or tensile strength.

In general, it can be summarized that PETG plastic reacted best to weathering. For this plastic, in most cases there was a statistically significant increase in hardness and an increase in tensile strength. A statistically significant decrease was not recorded for any of the effects. On the contrary, PLA plastic reacted the worst to weathering, with a decrease in hardness in most cases. The tensile strength of this material has increased in some cases and decreased in others. For ABS plastic, in most cases, the tensile strength limit was reduced, and the hardness, on the contrary, increased. For the ASA plastic, there was an “on average” increase in hardness. The ultimate tensile strength was not much affected.

### 6.3. Evaluation of Fracture Surfaces

From the observation of the nature of the fracture after the tensile test, it is evident that a brittle fracture occurs with a very small shape deformation in the case of ABS material; see Figure 12. The samples, after being placed in the freezer and condensation chamber, fractured without any shape deformation when the fracture took place in one plane. For samples that were placed in outdoor conditions for 3 months, the fracture was mixed with a small ductile deformation. Overall, it can be concluded that the fracture surfaces are very similar. From the detail of the fracture surface, it can be seen that during printing there are a very large number of holes on the edges of part of the samples, which are caused by the printing conditions.

From the nature of the fracture in the PETG material (see Figure 13), it is evident that this is the most flexible material of all. Not all PETG specimens ruptured, but the tensile test was set up so that the test was terminated when the Fmax dropped by more than 80%. At the same time, due to the shape of the fracture, the details of the fracture surfaces were not taken, because the fracture surfaces are very fragmented and the depth of field of the optical microscopes did not allow taking high-quality images. It is evident from the results that UV radiation had the highest effect when the color lightened and the sample was completely broken. In the remaining samples, it is possible to observe a very large elastic-plastic deformation, which is most pronounced in the sample after placement in the condensation chamber.

The fracture surfaces of the PLA material, which should be most susceptible to changes in properties after being placed in a degrading environment, are shown in Figure 14. From the shape of the fractures, it is clear that in the reference sample the fracture goes in the direction of the printing of the fibers and has a fractured cleavage shape. After exposure to the degradation effect, the fracture changes to planes, where the fracture goes across the surface of the specimen and is brittle with a minimal proportion of ductile fracture. When observing the fracture surfaces, it is evident how many large imperfections and gaps were created when printing the perimeters of samples. Compared to the ABS sample, the number of these imperfections is higher.

Fracture surfaces of ASA material samples (see Figure 15) show that this material shows the smallest dispersion in fracture shape after being in a degrading environment. In most cases, the fracture shape showed ductile failure; only in the case of UV 100 h was the fracture brittle with no indication of ductile failure. When analyzing the surface of the quarries, an imperfection can be seen again between the perimeter of the samples and the filling, where a large number of voids and imperfections can be observed.

The expected improving trend of mechanical properties was confirmed after placing the samples in the furnace at a temperature of 60 °C. For some of the materials, the strength limit was increased; for others, the value was not affected. At the same time, the PLA material is the only one that did not show a decrease in ductility. Hardness was not significantly affected. By comparing the results after placement in the condensation chamber (at 55 °C and 100% humidity) with the results after placing the samples in the furnace at a temperature of 60 °C, it is also possible to estimate the influence of humidity—a lower tensile strength limit is evident. The influence of humidity did not significantly affect ductility and hardness. There was no significant decrease in the mechanical properties after the samples were exposed to the weather. No significant influence can be observed on the ultimate strength values. A slight decrease was noted in the case of ductility, especially for PETG and ABS materials. Hardness was not significantly affected again.

## 7. Conclusions

In the article, the most significant influences that cause aging and degradation of polymers—light, temperature, oxygen, and water—were experimentally tested. When using a plastic part, the material is not stressed by isolated factors but by a combination of various harmful factors, e.g., simultaneous exposure to oxygen and light, heat, and mechanical stress. Solar radiation, especially UV radiation, has the greatest influence on the degradation of polymers. Ultraviolet rays break the bond between two atoms in the macromolecule chain, and the macromolecule breaks up into smaller parts (usually a radical is formed at the tertiary carbon), which then easily react with atmospheric oxygen to form aldehyde and carboxyl groups [45].

From the evaluation of the mechanical properties of plastic parts that were exposed to degradation effects, it was found that these effects do not have a fundamental (unequivocally demonstrable) effect on the mechanical properties of the part evaluated during short-term loading. However, this statement cannot be generalized, as the change in the properties of the material due to its aging and degradation is influenced by the addition of the polymer (type and amount of stabilizers), which is the know-how of each manufacturer and not published anywhere. This degradation testing is therefore necessary, especially in the field of increasingly widespread hobby FFM printing, where, although the manufacturer sets the values of the mechanical properties in the technical sheet, the values differ for each batch of material and for each color. Based on the knowledge gained, it seems more appropriate to subject plastics to long-term cyclic loading, which can be more sensitive to structural changes of the material due to its aging, in contrast to normal static loading.

From the point of view of ultimate strength values, PLA appears to be the best material, but it is necessary to take into account the negative influence of the condensation chamber, temperature cycles, and weather effects. It can be assumed that this decrease will be more significant as the duration of these effects increases. After placing the PLA samples in the furnace at a temperature of 60 °C, there was no deterioration of the properties, but this was a threshold temperature since the use at higher temperatures is limited by the softening temperature and the related shape stability. PLA also achieved the highest hardness and, conversely, showed the lowest ductility. For the reasons described above, the PETG material can be evaluated as more suitable in terms of tensile strength. Although the measured values are slightly lower than in the case of PLA, there was no decrease in the ultimate tensile strength for any of the factors—on the contrary, a slight increase was observed for all factors. For that reason, PETG material can be considered suitable for use in these environments in terms of strength. The disadvantage of PETG material compared to other materials is its lower hardness. Although there was a significant drop in ductility due to all factors, PETG still achieved the highest ductility of all tested materials. The ABS material reached the lowest values of the tensile strength limit, which was also observed to decrease due to all factors; therefore, the material cannot be recommended based on the results compared to the others tested.

In the case of ASA material, the properties were least affected by individual factors. Although ASA achieved lower strength limit values compared to PLA and PETG, its advantage is precisely the stability of all monitored properties. Compared to PETG, it achieved higher hardness, and at the same time, according to the analysis in the theoretical part, the ASA material resists higher temperatures compared to PETG. For these reasons, its use can also be recommended in all tested environments.

## Figures and Tables

**Figure 1 materials-16-03268-f001:**
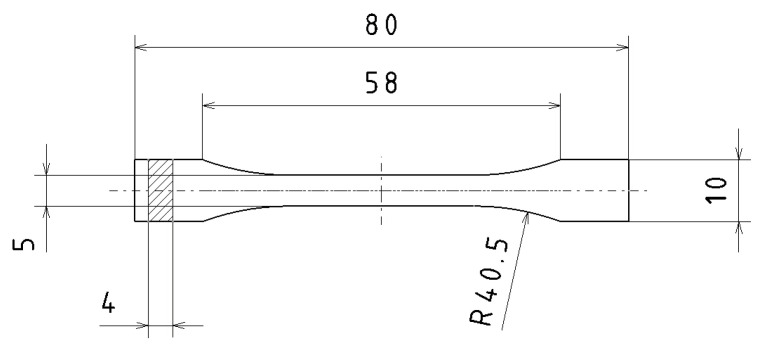
Schematic representation of the selected test rod.

**Figure 2 materials-16-03268-f002:**
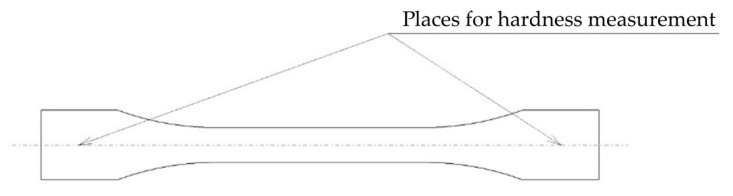
Places for hardness measurement.

**Figure 3 materials-16-03268-f003:**
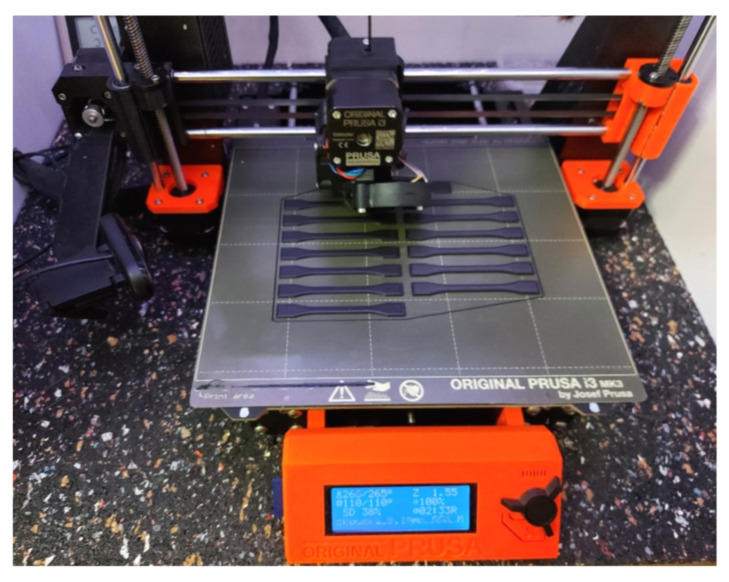
Preparation of samples on a 3D printer.

**Figure 4 materials-16-03268-f004:**
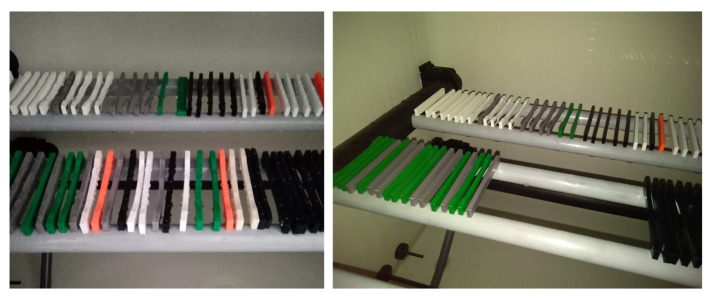
Set of test samples for placement in the condensation chamber.

**Figure 5 materials-16-03268-f005:**
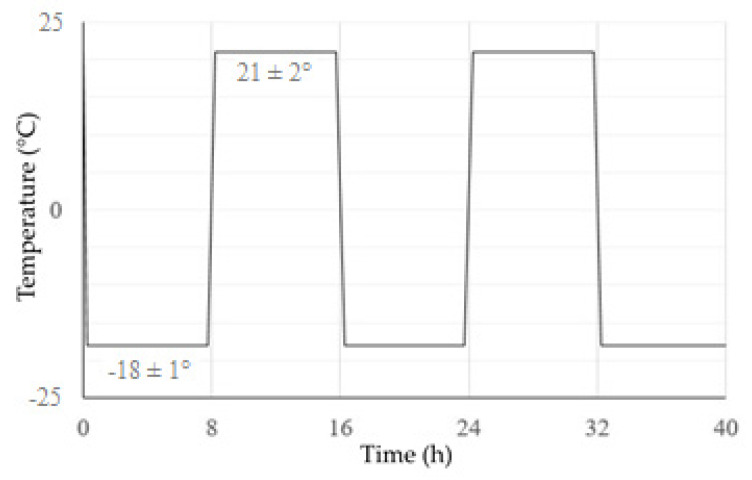
Schematic of the temperature cycling process.

**Figure 6 materials-16-03268-f006:**
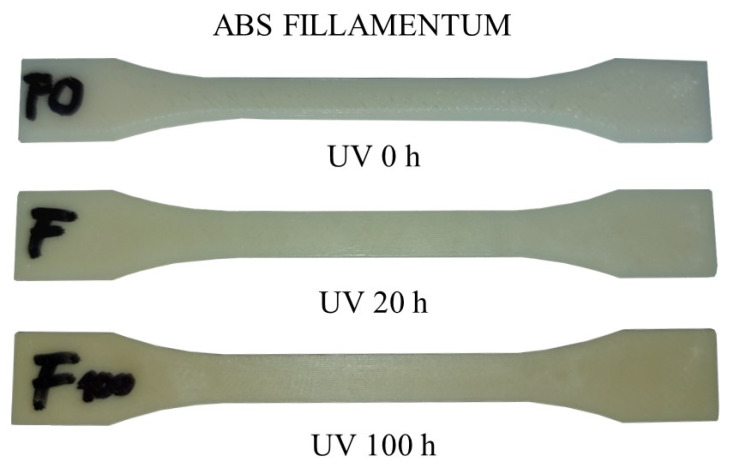
Color change due to UV radiation—Fillamentum ABS.

**Figure 7 materials-16-03268-f007:**
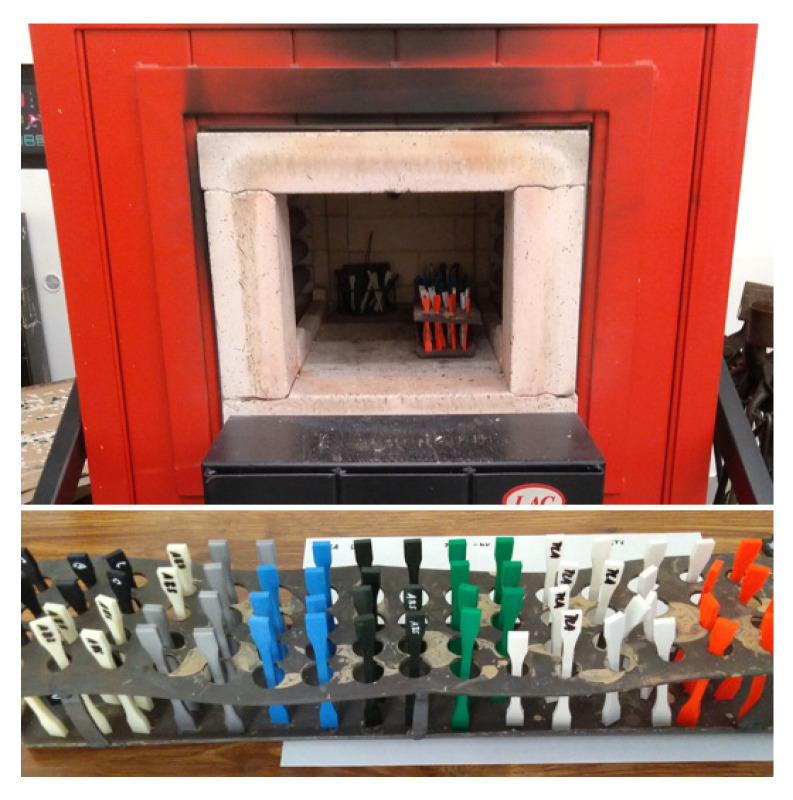
Set of test samples for placing in the furnace.

**Figure 8 materials-16-03268-f008:**
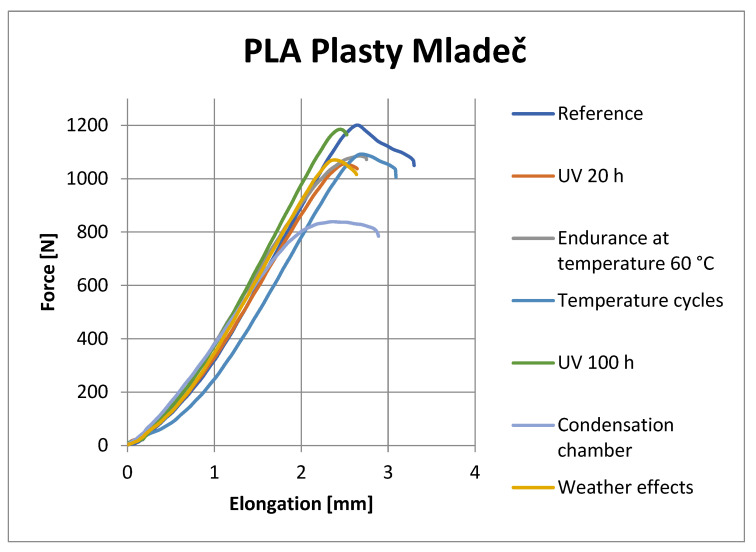
Tensile curves of PLA Plasty Mladeč samples.

**Figure 9 materials-16-03268-f009:**
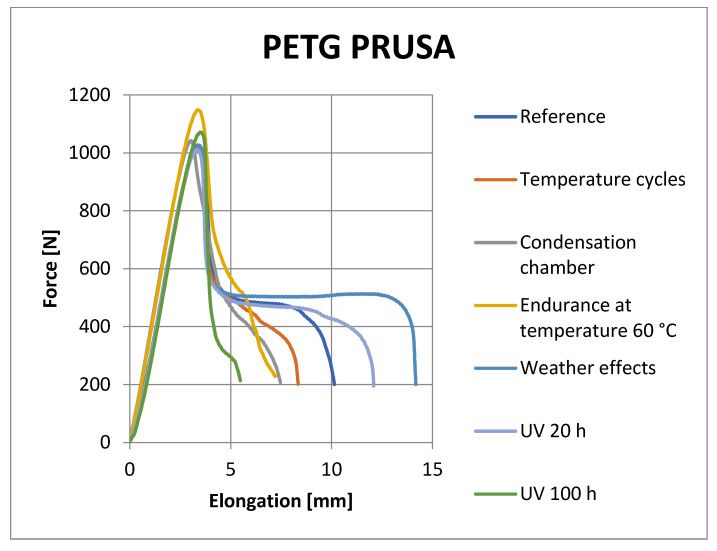
Tensile curves of PETG Prusament samples.

**Figure 10 materials-16-03268-f010:**
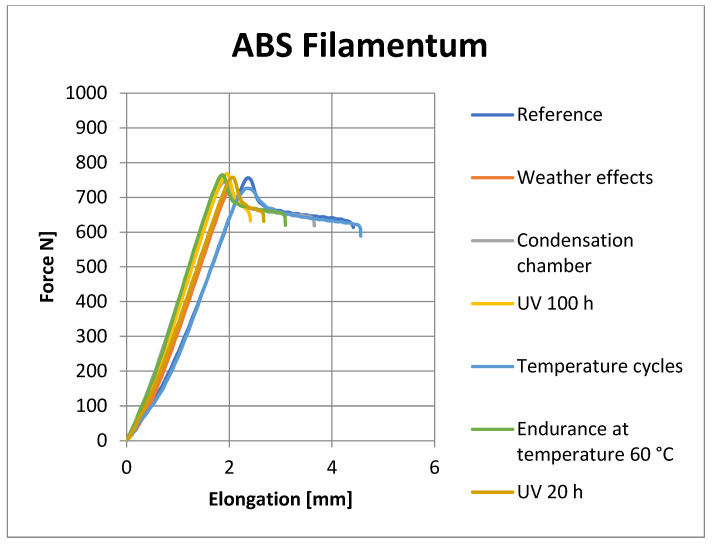
Tensile curves of ABS Fillamentum samples.

**Figure 11 materials-16-03268-f011:**
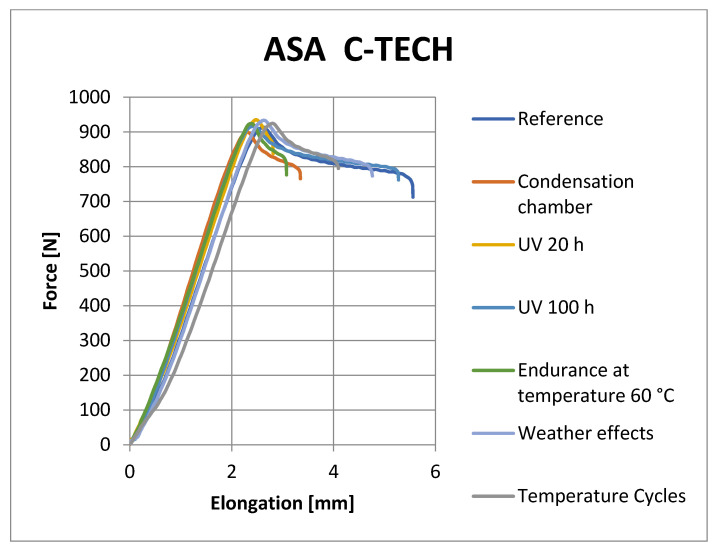
Tensile curves of ASA C-TECH samples.

**Figure 12 materials-16-03268-f012:**
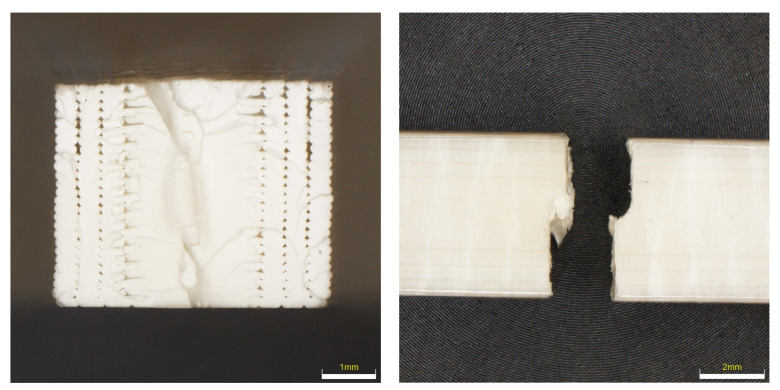
Character of fracture after tensile testing for ABS material.

**Figure 13 materials-16-03268-f013:**
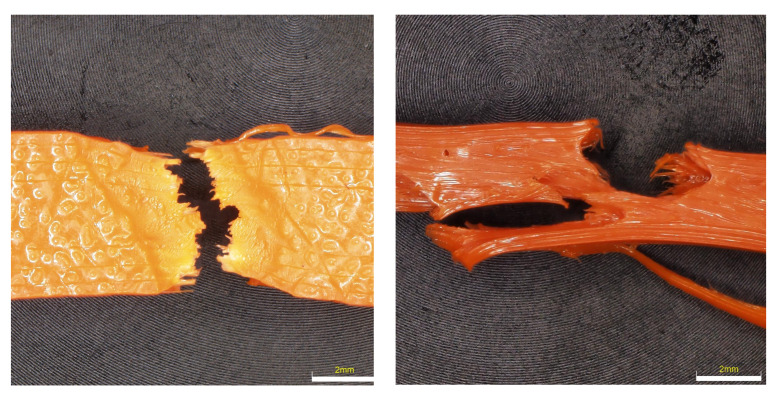
Character of fracture after tensile testing for PETG material.

**Figure 14 materials-16-03268-f014:**
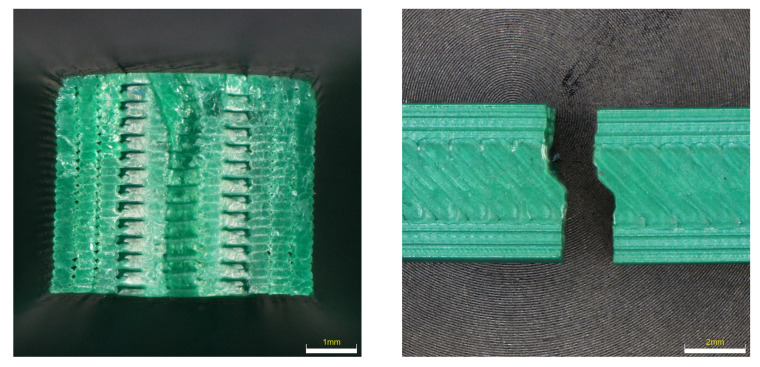
Character of fracture after tensile testing for PLA material.

**Figure 15 materials-16-03268-f015:**
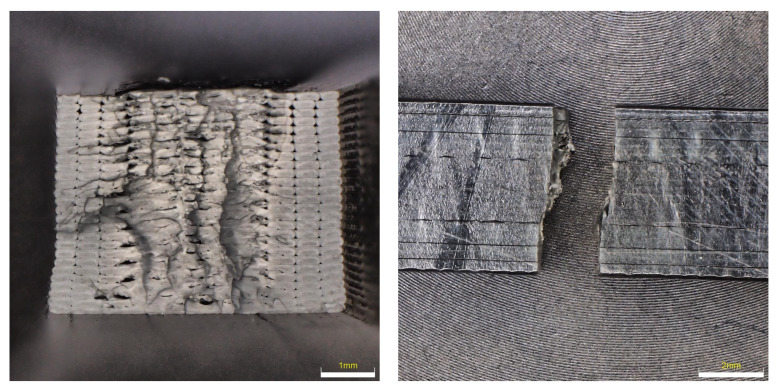
Character of fracture after tensile testing for ASA material.

**Table 1 materials-16-03268-t001:** Overview of materials used and individual manufacturers.

Material	Producer	Color	Material	Producer	Color
PLA	Prusament	Galaxy Silver	ABS	Fillamentum	Natural
	Plasty Mladeč	Green		Gembird	Black
	Spectrum	White Polar		Plasty Mladeč	Blue
PETG	Prusament	Orange	ASA	C-TECH	Black
	Plasty Mladeč	Black		Devil Design	Black
	Spectrum	White Arctic		Plasty Mladeč	Natural

**Table 2 materials-16-03268-t002:** Selected properties of applied additive materials [38,39,40].

Variable	Unit	PLA	ABS	PETG	ASA
Ultimate tensile strength	(MPa)	60	40	53	55
Impact strength Charpy	$\left(kJ/m^{2}\right)$	16.5	28	52	34.5
Nozzle temperature	(°C)	210–220	240–255	230–250	255–265
Glass transition temperature	(°C)	60	100	81	110
Hardness Shore D	(HShD)	80	75	72	75

**Table 3 materials-16-03268-t003:** Printing temperatures and speeds of selected materials [38,39,40].

Material	PLA	ABS	PETG	ASA
Bed temperature [°C]	60	110	85	110
Extruder temperature [°C]	220	240	240	250
Print speed for perimeters [mm/s]	35	40	45	30
Print speed for infill [mm/s]	80	80	80	80

**Table 4 materials-16-03268-t004:** The average increases in weight of individual materials after 100 h in the condensation chamber.

Material	Producer	Mass Increase	Material	Producer	Mass Increase
PLA	Prusament	2.87%	ABS	Fillamentum	1.13%
	Plasty Mladeč	1.70%		Gembird	0.87%
	Spectrum	3.39%		Plasty Mladeč	0.69%
PETG	Prusament	1.02%	ASA	C-TECH	1.18%
	Plasty Mladeč	0.43%		Devil Design	0.91%
	Spectrum	1.13%		Plasty Mladeč	0.84%

**Table 5 materials-16-03268-t005:** Measured results—PLA material.

	PLA	Reference Sample	UV 20 h	UV 100 h	Condensation Chamber	Temperature Cycles	Endurance at Temperature 60 °C	Weather Effects
Ultimate tensile strength (MPa)	Plasty Mladeč	59.10 ± 0.68	54.90 ± 0.87	58.80 ± 0.86	42.60 ± 0.70	56.70 ± 0.90	54.20 ± 0.96	55.80 ± 0.95
	Spectrum	51.30 ± 0.75	51.00 ± 1.06	59.80 ± 0.90	50.10 ± 0.96	53.10 ± 0.96	58.10 ± 0.94	51.60 ± 0.79
	Prusament	52.50 ± 0.87	51.10 ± 1.07	56.00 ± 0.92	40.50 ± 0.78	50.90 ± 0.78	54.20 ± 0.88	52.70 ± 0.80
Hardness (HShD)	Plasty Mladeč	80.61 ± 0.71	79.29 ± 0.85	80.65 ± 0.82	78.20 ± 0.63	80.58 ± 0.82	76.50 ± 0.91	81.17 ± 0.92
	Spectrum	78.16 ± 0.73	77.46 ± 0.93	80.31 ± 0.81	73.40 ± 0.92	78.11 ± 0.95	78.13 ± 0.93	76.29 ± 0.74
	Prusament	80.59 ± 0.82	77.99 ± 0.93	81.59 ± 0.88	74.96 ± 0.74	79.20 ± 0.72	77.18 ± 0.81	80.30 ± 0.76

**Table 6 materials-16-03268-t006:** Measured results—PETG material.

	PETG	Reference Sample	UV 20 h	UV 100 h	Condensation Chamber	Temperature Cycles	Endurance at Temperature 60 °C	Weather Effects
Ultimate tensile strength (MPa)	Plasty Mladeč	49.90 ± 0.65	51.90 ± 0.78	54.30 ± 0.96	52.50 ± 0.58	51.70 ± 0.87	56.40 ± 0.69	52.50 ± 0.75
	Spectrum	49.50 ± 0.69	51.30 ± 0.94	54.30 ± 0.85	53.40 ± 0.69	50.40 ± 0.74	57.50 ± 0.62	50.70 ± 0.72
	Prusament	49.90 ± 0.83	50.60 ± 0.74	54.20 ± 0.68	52.20 ± 0.71	51.40 ± 0.68	56.80 ± 0.86	50.80 ± 0.84
Hardness (HShD)	Plasty Mladeč	72.82 ± 0.61	75.87 ± 0.72	75.15 ± 0.91	73.12 ± 0.52	73.96 ± 0.89	73.18 ± 0.75	75.29 ± 0.79
	Spectrum	73.64 ± 0.74	73.99 ± 0.89	72.98 ± 0.81	73.83 ± 0.73	73.71 ± 0.79	75.98 ± 0.68	74.58 ± 0.78
	Prusament	73.82 ± 0.87	74.62 ± 0.77	73.27 ± 0.75	70.12 ± 0.81	73.44 ± 0.77	72.66 ± 0.88	74.48 ± 0.81

**Table 7 materials-16-03268-t007:** Measured results—ABS material.

	ABS	Reference Sample	UV 20 h	UV 100 h	Codensation Chamber	Temperature Cycles	Endurance at Temperature 60 °C	Weather Effects
Ultimate tensile strength (MPa)	Plasty Mladeč	37.20 ± 0.65	37.10 ± 0.78	36.60 ± 0.96	37.30 ± 0.58	36.00 ± 0.87	36.60 ± 0.69	36.90 ± 0.75
	Fillamentum	38.50 ± 0.74	38.01 ± 0.87	37.60 ± 0.69	37.20 ± 0.71	36.30 ± 0.96	37.80 ± 0.76	36.00 ± 0.88
	Gembird	45.20 ± 0.75	45.60 ± 0.87	44.80 ± 0.69	45.00 ± 0.87	44.10 ± 0.71	45.10 ± 0.69	42.10 ± 0.81
Hardness (HShD)	Plasty Mladeč	75.07 ± 0.69	75.84 ± 0.74	77.68 ± 0.91	73.98 ± 0.63	74.40 ± 0.81	73.58 ± 0.64	77.07 ± 0.72
	Fillamentum	74.01 ± 0.78	76.64 ± 0.85	76.07 ± 0.64	75.26 ± 0.75	73.80 ± 0.91	74.20 ± 0.72	77.69 ± 0.84
	Gembird	76.47 ± 0.71	79.37 ± 0.82	80.35 ± 0.66	76.74 ± 0.84	75.73 ± 0.77	77.10 ± 0.64	79.25 ± 0.83

**Table 8 materials-16-03268-t008:** Measured results—ASA material.

	ASA	Reference Sample	UV 20 h	UV 100 h	Condensation Chamber	Temperature Cycles	Endurance at Temperature 60 °C	Weather Effects
Ultimate tensile strength (MPa)	Plasty Mladeč	46.20 ± 0.85	46.50 ± 0.69	46.20 ± 0.74	44.20 ± 0.85	46.40 ± 0.78	44.30 ± 0.64	47.00 ± 0.72
	C-TECH	44.90 ± 0.74	45.30 ± 0.68	44.20 ± 0.71	44.40 ± 0.86	45.70 ± 0.74	44.70 ± 0.76	46.30 ± 0.84
	Devil Design	45.30 ± 0.95	45.00 ± 0.94	44.00 ± 0.85	45.40 ± 0.64	45.90 ± 0.72	45.80 ± 0.83	46.30 ± 0.81
Hardness (HShD)	Plasty Mladeč	75.40 ± 0.81	77.23 ± 0.68	77.11 ± 0.73	76.12 ± 0.81	75.21 ± 0.73	73.66 ± 0.71	76.93 ± 0.75
	C-TECH	74.45 ± 0.77	77.26 ± 0.65	76.68 ± 0.73	75.30 ± 0.87	73.82 ± 0.75	73.56 ± 0.71	75.65 ± 0.81
	Devil Design	75.76 ± 0.92	76.41 ± 0.86	76.63 ± 0.81	75.78 ± 0.75	75.78 ± 0.78	74.84 ± 0.81	76.44 ± 0.88

## Data Availability

Not applicable.

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
