# Peer review of "Analysis of the Mechanical Properties of 3D-Printed Plastic Samples Subjected to Selected Degradation Effects"

_materials, 2023, doi:10.3390/ma16083268_

Round 1
Reviewer 1 Report
Authors of the publication: "Analysis of mechanical properties of 3D printed plastic samples subjected to selected degradation effects" presented the results on the mechanical properties (Rm, hardness) of the selected materials (PLA, PETG, ABS, ASA) tested in the non-degenerate state and after exposure of the samples to the selected degradation factors. The authors included degradation factors as: UV radiation, high temperature environments, high humidity environments, temperature cycles and exposure to weather conditions were monitored.
The topic discussed in the article is very interesting and very important in the development of materials used in 3D printers and determining the life of printed elements.
However, the work has a number of errors. For example:
1. What is the difference between FFF and FDM methods? As for me, it is certainly not related to the heated table or chamber.
2. Did each manufacturer of the selected material have the same characteristics (ultimate tensile strength, impact strength Charpt, etc.)
3. Lack of 3D printing characteristics, e.g. extruder temperatures, heatbed temperatures, nozzle size, filament diameter, single line width, print speed, etc.
4. Were the printed samples within the dimensional tolerance?
5. Fig. 14 and 15 - the results are in the tables. These drawings should be removed.
6. Figures 16 - 19 - unacceptable quality of photos.
Author Response
Dear Sir/Madam, Thank you for your valuable time spent reviewing our article. Your comments have been incorporated into the article to the best of our knowledge and are highlighted in yellow. Thanks again for helping to improve our article.
Please see the attachment

Reviewer 2 Report
The paper presents a work on mechanical properties of 3D printed plastic samples under degradation. The topic is interesting and the manuscript is well-written. I suggest publication after addressing the remarks given below.
· Please check the text for typos and grammar.
· Pg.1 ln.34: …FDM (Fused Deposition Modeling)… Please use the abbreviation in the parenthesis.
· Novelty in this work should be given clearly.
· Is there any standard followed in the degradation tests?
· How did authors determine the limits/conditions in the degradation tests?
· Why not FTIR used in the investigations? Maybe after UV exposure? Authors could benefit from the paper below in UV degradation.
o https://doi.org/10.3390/ma15093269
o https://doi.org/10.1016/j.carbpol.2019.04.062
· Analyses are macro-level. Micrography could be used.
· Fig.14 and 15 are not easily understandable. Please give the results in a better way.
Author Response

(The authors gave the same response as above.)

Reviewer 3 Report
The manuscript titled „Analysis of mechanical properties of 3D printed plastic samples subjected to selected degradation effects” was reviewed.
Paper is well written and it was clearly explained for reader stand points. The Authors have provided a broad understanding of the topic.
What the paper lacks in my opinion is a decent discussion on the nature of the changes observed in the materials that were subjected to the different degradation mechanisms. Most of the discussion covered in the article sticks to the “what can be observed on the presented graphs” type of results interpretation. In my opinion this is not enough for an article which is supposed to be published in “Materials”. Please elaborate the changes observed during your investigations from a material degradation point of view i.e. the major chemical changes (oxidation and chain scission), leading to reduction in the molecular weight, degree of polymerization of the polymers maybe? What is the influence of different additives used by the filament producers?
Language is fine, although what caught my attention was row 335 “After the action of the "UV Lamp"”. Please rephrase that.
Graphs are also fine, except Fig. 14 and 15 which are very hard to read. Can you please change the form of the graphs mentioned or perhaps use a table given data instead?
Author Response
Dear Sir/Madam, Thank you for your valuable time spent reviewing our article. Your comments have been incorporated into the article to the best of our knowledge and are highlighted in yellow. Thanks again for helping to improve our article.

Round 2
Reviewer 3 Report
The revised version of the manuscript is a one small step better in comparison to the previous one. In my opinion it still would require a more deeper discussion regarding the changes in the evaluated materials exposed to different degradation types. However as I am aware of the difficulties in obtaining the necessary data from the filament producers we can assume that this research will indeed be helpful for the 3D printing community when choosing the right material for a specific outdoor application, without investigating too much into the materials structure.
After a second reading I would suggest some language improvements. I.e. row 332 “at the case” – I believe it was supposed to be “in the case”, this is repeated in 336 and 414. Row 341, 342 “in [3] states”, “ in [44, 45] he states” – please consider rephrasing. Row 373 “in the packaging part of the samples” – I haven’t found any explanation what is a packaging part of the sample, I can only assume that you mean the external shell or something similar.